# Thai Native Chicken as a Potential Functional Meat Source Rich in Anserine, Anserine/Carnosine, and Antioxidant Substances

**DOI:** 10.3390/ani11030902

**Published:** 2021-03-22

**Authors:** Sukanya Charoensin, Banyat Laopaiboon, Wuttigrai Boonkum, Jutarop Phetcharaburanin, Myra O. Villareal, Hiroko Isoda, Monchai Duangjinda

**Affiliations:** 1Department of Animal Science, Faculty of Agricultural, Khon Kaen University, Khon Kaen 40002, Thailand; sukanya_charoensin@kkumail.com (S.C.); banyat@kku.ac.th (B.L.); wboonkum@gmail.com (W.B.); 2Network Center for Animal Breeding and Omics Research, Faculty of Agricultural, Khon Kaen University, Khon Kaen 40000, Thailand; 3Department of Biochemistry, Faculty of Medicine, Khon Kaen University, Khon Kaen 40002, Thailand; jutarop@kku.ac.th; 4Faculty of Life and Environmental Sciences, University of Tsukuba, Tsukuba City 305-8572, Japan; villareal.myra.o.gn@u.tsukuba.ac.jp; 5Alliance for Research on North Africa (ARENA), University of Tsukuba, Tsukuba City 305-8572, Japan

**Keywords:** anserine, carnosine, functional meat, Thai native chicken, antioxidant

## Abstract

**Simple Summary:**

Potential bioactive compounds are properties that can play an important role in the prevention of chronic diseases, exhibit neuroprotective effects, and have antioxidant, anti-aging, and antiglycation properties. Anserine (β-alanyl-3-methyl-L-histidine) and carnosine (β-alanyl-L-histidine), dipeptides found in native chicken meat, are good sources of antioxidant substances. Anserine is a biomarker in the muscle of chickens, making native chicken meat a source of antioxidant compounds. However, there are no studies on bioactive compounds in Thai native chickens. This study was conducted to identify the presence of dipeptide anserine in Thai native and Thai native crossbred chicken meat using nuclear magnetic resonance spectroscopy and to determine the antioxidant activity of chicken breast extract. We found that Thai native chicken was rich in anserine, anserine/carnosine, and antioxidant substances; therefore, Thai native chicken might have the potential to be a functional meat source.

**Abstract:**

This study identified anserine and anserine/carnosine in chicken breast of Thai native chicken (TNC; 100% Thai native), Thai synthetic chicken (TSC; 50% Thai native), and Thai native crossbred chicken (TNC crossbred; 25% Thai native) compared with commercial broiler chicken (BR; 0% Thai native) using nuclear magnetic resonance (NMR) spectroscopy and the effect on antioxidant activity using 2,2-diphenyl-1-picrylhydrazyl assay (DPPH). We conducted experiments with a completely randomized design and explored principal components analysis (PCA) and orthogonal projection to latent structure-discriminant analysis (OPLS-DA) to identify the distinguishing metabolites and relative concentrations from ^1^H NMR spectra among the groups. The relative concentrations and antioxidant properties among the groups were analyzed by analysis of variance (ANOVA) using the general linear model (GLM). This study revealed seven metabolites alanine, inositol monophosphate (IMP), inosine, and anserine/carnosine, lactate, anserine, and creatine. Lactate, anserine, and creatine were major components. In terms of PCA, the plots can distinguish BR from other groups. OPLS-DA revealed that anserine and anserine/carnosine in the chicken breast were significantly higher in TNC, TSC, and TNC crossbred than BR according to their relative concentrations and antioxidant properties (*p* < 0.01). Therefore, TNCs and their crossbreeds might have the potential to be functional meat sources.

## 1. Introduction

In the past few years, there has been a breeding goal of native chicken improvement in Asian countries, including Thailand; with this goal, chicken breeding has focused on morphological selection, physical properties, and production performance [1,2,3]. Recently, global consumers have been interested in natural food or a food that contains special quality nutrients/bioactive compounds that are beneficial to their health and wellbeing, known as functional food [4,5]; consumers value these traits because they are concerned that excess food consumption (overnutrition) is linked to a variety of diseases. The properties of potential bioactive compounds play an important role in neuroprotective, antioxidant, anti-aging, and antiglycation properties, as well as the prevention of chronic diseases [6,7].

Native chicken is an important possible source of functional food because it contains considerable amounts of bioactive compounds when compared with commercial broiler chicken and other meat, including angiotensin-converting enzyme inhibitors (ACE-I inhibitor), L-carnitine, creatine, carnosine, and anserine [8,9,10]. The dipeptides anserine (β-alanyl-3-methyl-L-histidine) and carnosine (β-alanyl-L-histidine) are good sources of antioxidant substances and serve as biomarkers in the muscle of chickens [9,11,12]; interestingly, their levels are higher in native chickens compared with commercial broilers and other meat [9,10]. In addition, anserine, also available for antifatigue with balancing of lactic acid, represents a physiological buffer in skeletal muscles [13,14,15].

From the above, Thai native chickens (TNCs) might have the potential to be a functional chicken meat. These chickens are an important genetic resource for Thailand. A few years ago, performance improvements in TNC were achieved in terms of growth and egg production by the Research and Development Network Center for Animal Breeding (Native Chicken). Pradu Hang Dam Mor Kor 55 (PD) exhibited excellent growth performance, whereas Chee KKU 12 (CH) was superior in egg production. Consequently, both breeds are popular, and PD has been used as a purebred animal, especially in the northeast of Thailand, because of its excellent appearance, morphological features, and growth rate. In terms of CH, in this breed, researchers have been working on accelerating growth and improving egg production to achieving have developed Thai synthetic chicken (TSCs) which are sustainable with TNC utilization to retain the native genetic fraction 50% and have developed a terminal cross to retained the native genetic fraction 25% known as KKU-ONE. Although these birds are being used for growth and egg performance, their meat quality potential has not been fully exploited, despite the high importance of this aspect for current consumers. From an animal breeding viewpoint, identification of bioactive compounds as new traits is interesting for building up a new breeding goal that follows recent consumer trends. According to many enterprises, it is interesting to try to take TNC and their crossbred up on modern trade, which requires the products contain quality nutrients/bioactive compounds that are beneficial to human health and wellbeing.

In the past, the perception and promotion of Thai native chicken meat have related to production performance, physical properties, such as the characteristics of muscle fibers and shear force [16,17], which are responsible for the meat’s unique texture. Regarding biochemical properties, chicken meat contains high levels of collagen, protein, and inositol monophosphate (IMP) [18,19]. However, there is no information about the potential bioactive compounds in TNC and their crossbred counterparts. Therefore, this study focused on the identification of essential elements/active ingredients/bioactive in TNC and their crossbred to finding a new trait that is useful for setting up a further new breeding goal. Moreover, to increase community income in a sustainable way by developing and promoting functional food and products for an aging society, the findings are expected to open opportunities for enhancing the competitiveness of the healthy food market. The value added to natural meat or products is determined based on a study of chemical properties and compared among native and crossbred chickens, in relation to commercial chickens, using advanced techniques.

One alternative is the use of nuclear magnetic resonance (NMR) spectroscopy, which has advanced beyond conventional methods because it is a rapid method to identify all metabolite profiles; it has been successfully applied to the analysis of chicken meat quality at the metabolite level [20]. Moreover, some of the special properties of NMR are that it is non-destructive, involves non-targeted analysis, provides high-throughput data, and directly measures components at the molecular level, requiring only a single standard to quantify several components simultaneously [21,22]. Therefore, this study was conducted to identify the presence of dipeptide anserine in chicken breast extract of Thai native and crossbred chickens using NMR spectroscopy and to determine the antioxidant activity of chicken breast extract.

## 2. Materials and Methods

### 2.1. Animal Raising and Tissue Collection

The experiment was approved by the Institutional Animal Care and Use Committee of Khon Kaen University (IACUC-KKU-58/62). A total of 500 birds from two breeds of TNC (100% Thai native), PD and CH; Thai synthetic chicken (TSC; 50% Thai native), Khai Mook E-san (KM), and Thai native crossbred chicken (TNC crossbred; 25% Thai native), KKU-ONE. We used 100 day-old chicks per breed from the Research and Development Network Center for Animal Breeding of Khon Kaen University. Except for commercial broiler chicken, we used day-old chicks from Arbor Acer, part of the Charoen Pokphand Company, which were assigned to a completely randomized design with four replications within the breeds. All chickens were raised under the same environmental conditions, with open-air housing and a vaccination program. Feed was provided *ad libitum* with a commercial broiler diet; first, a starter feed containing 21% crude protein (CP), 3100 kcal of ME/kg, and 5% crude fiber was given to chicks aged 1–3 weeks; then, for the growing period, second, the feed contained 19% CP and 3200 kcal of ME/kg, and this was fed at 4 weeks of age until slaughtering. Both feeds consisted of soybean meal, bone meal, mineral, amino acids, and so on in the formula. However, there were no details on the amount and kind of amino acids in these commercial diets.

Fifty chicken breasts (*pectoralis major*) from five males and five females of each breed were sampled at marketing age at 8 (TNC crossbred), 10 (TSC), and 12 (TNC) weeks old, following the local market of Thailand, Thai consumer usual consumed approximately 1.2–1.6 kg of chicken weight [23]. Commercial broiler chickens were sampled at 6 weeks old following the retail cut market in Thai supermarkets. Fifty chicken breasts were slaughtered according to Jaturasitha et al. [17], with some modifications. We collected the individual left chicken breasts and stored them at −20 °C in a sealed bag until analysis with NMR spectroscopy and antioxidant assay using DPPH.

### 2.2. NMR Spectroscopy Assay

#### 2.2.1. Chicken Metabolite Extraction

A total of 50 chicken breasts (*pectoralis major*) were processed; briefly, 300 mg of minced chicken breast was transferred into 15 mL centrifuge tubes and extracted according to Fathi et al. [24] with some adjustments. After the addition of 4 mL of methanol and 0.85 mL of purified water (high-performance liquid chromatography [HPLC] grade) per gram of chicken tissue, the samples were vortexed. Subsequently, we added another 2 mL of chloroform per 1 g of sample, followed by vortexing. Next, another 2 mL of chloroform and purified water were added per gram of the sample, and the mixture was placed on ice for 15 min. The samples were vortexed again and centrifuged at 1000× *g* for 15 min at 4 °C. The extracted samples were separated into two phases—aqueous extract samples and the upper methanol/water phase (polar metabolites)—then transferred into a micro-centrifuge. After removing the solvents with a speed vacuum concentrator (CentriVap Concentrator, Labconco, Kansas City, MO, USA), the samples were kept at 80 °C until NMR analysis and antioxidant assay by DPPH.

#### 2.2.2. ^1^H NMR Metabolic Profiling, Data Pre-Processing, and Metabolite Identification

The aqueous extracts were re-suspended with 580 µL of 100 mM sodium phosphate buffer in D_2_O containing 0.1 mM 3-trimethysilypropionic acid (TSP; Cambridge Isotype Laboratories, Cambridge, MA, USA) and 0.2% NaN_3_ at pH 7.4. The mixture was vortexed and centrifuged at 12,000× *g* for 1 min. Subsequently, 550 mL of the supernatant was transferred into an NMR tube with a diameter of 5 mm. Untargeted ^1^H NMR metabolomic profiling was conducted, and spectra were recorded using a 400 MHz NMR spectrometer with a CryoProbe (Bruker, Billerica, MA, USA). All samples were detected in a standard 1-dimension pulse sequence (recycle delay-90°-t1-90°-tm-°-acquisition) with t1 and 3 ms, tm to 10 ms, and 90° pulse to 10 µs in 64 scans. Chemical shift referencing, baseline correction, and phasing were performed using TopSpin (version 4.0) software to adjust the peak alignment, normalization, and scaling. The water peak was excised from the tissue spectra to minimize the effect of the remaining baseline distortion caused by imperfect water suppression. To confirm the assignment of correlated resonances, statistical total correlation spectroscopy (STOCSY) was employed [25]. Moreover, the resonances of interest were searched against online metabolite databases, such as the Biological Magnetic Resonance DataBank and the Human Metabolome Database.

#### 2.2.3. Multivariate Statistical Analysis

A processed spectral data matrix was employed to conduct principal component analysis (PCA) with a unit variance (uv) scaling method to visualize metabolic similarities and differences and to identify possible outliers, followed by an orthogonal signal correction–projection to latent structures–discriminant analysis (OPLS-DA) in a MATLAB R2015a (MathWorks, Natick, MA, USA) environment. OPLS-DA scores and coefficient plots were generated, with a color visualization of the correlation values |r| of each variable. Red indicates higher correlation, whereas blue indicates lower correlation of the variables with the classification. The fitness and predictability of the models obtained from the OPLS-DA were determined by the R2 and Q2 values, respectively. The model validity was determined by the permutation *p*-value of each model. The OPLS-DA models in the current study were constructed based on one PLS component and one orthogonal component using mean-centered and uv-scaled spectral datasets. The validation of all OPLS-DA and OPLS models involved in this study was assessed using the permutation *p*-value (*p* < 0.05).

#### 2.2.4. Relative Concentrations of Anserine and Anserine/Carnosine Content

The area under the peak of anserine and anserine/carnosine was calculated as the relative concentration using MATLAB software (R 2015a). The least square means of relative concentrations among groups were analyzed via analysis of variance (ANOVA) using the general linear model (GLM) as follows: $Y_{ijkl}=Breed_{i}+Sex_{j}+\varepsilon_{ijkl}$, where $Y_{ijkl}$ is the relative concentration of anserine and anserine/carnosine; $Breed_{i}$ represents KKU-ONE, KM, CH, PD, and BR; $Sex_{j}$ is male or female; and $\varepsilon_{ijkl}$ is the experimental error by the Statistical Analysis System (SAS) statistical procedure (SAS, 2019).

### 2.3. Antioxidant Assay

#### 2.3.1. Breast Extracts Rich in Dipeptides (Anserine and Carnosine)

Using a total of 20 chicken breasts (two males and two females from each breed), the dipeptide-rich chicken extracts were obtained according to ref. [9] with some modifications. Briefly, 1 g of finely chopped chicken breast was transferred into a 15 mL centrifuge tube and homogenized after adding 10 mL of distilled water via a homogenizer (Witeg, Wertheim, Germany) at 25,000× *g* for 1 min. The homogenized sample was centrifuged at 4500× *g* and 4 °C for 15 min, and the supernatant was transferred and incubated at 80 °C for 10 min in a water bath. The aqueous extracts were filtrated through a 0.45 μm membrane (Filtrex, AMK, Singapore) after centrifugation at 4500× *g* for 30 min. The chicken extracts containing dipeptides (anserine and carnosine) were mixed sexes with four samples comprising two males and two females of each breed; this mixing because of previous results showing that interactions between breed and sex have no significance. They were stored at −20 °C until further analysis.

#### 2.3.2. Antioxidant Effects of Chicken Extracts Containing Dipeptides (Anserine and Carnosine)

The antioxidant activity of the chicken extracts was investigated using the DPPH assay. A DPPH working solution was prepared at a ratio of 5:1:2 (0.2 M 1,1-diphenyl-2-picrylhydrazyl; DPPH, 0.4 M MES, and MilliQ water, respectively). The chicken breast extracts containing dipeptides were prepared at 10^−1^, 10^−3^, 10^−5^, 10^−7^, and 10^−9^ dilutions using distilled water. The mixtures were incubated for 10 min at room temperature after adding the DPPH working solution (wrapped in aluminum foil), and the absorbance was measured spectrophotometrically at a wavelength of 520 nm and calculated using this equation:

The anti-oxidation ratio (%) was calculated by $\left(1-\frac{\left(\mathrm{Sample}-\mathrm{Blank}\right)}{\mathrm{Control}}\right)\times 100$. The antioxidants were analyzed by ANOVA using the GLM as follows: $Y_{ijkl}=Breed_{i}+Sex_{j}+\varepsilon_{ijkl}$, where $Y_{ijkl}$ is the antioxidant; $Breed_{i}$ represents KKU-ONE, KM, CH, PD, and BR; and $\varepsilon_{ijkl}$ is the experimental error by the SAS statistical procedure [26].

## 3. Results and Discussion

### 3.1. Global ^1^H-NMR Metabolic Profiling of Chicken Extract

Representative ^1^H-NMR spectra (0–10 ppm) obtained from the chicken breast extracts of TNC, TSC, TNC crossbreds, and commercial broilers exhibited seven major metabolites that are shown in Table 1 and Appendix A. These include lactate, alanine, anserine, anserine/carnosine, IMP, creatine, and inosine. Most of these metabolites have also been previously found in chicken meat [27,28]. In the current study, three metabolites—creatine, lactate, and anserine—were found to be most abundant, which is consistent with previous studies [20,27,28]. Creatine is a key metabolite associated with energy metabolism in muscles, and it is related to reduced accumulation of lactate in muscle; hence, it can potentially improve chicken meat quality [29]. Furthermore, lactate has been shown to be related to meat quality and to negatively contribute to the water-holding capacity, and consequently, meat tenderness [30]. Anserine is a dipeptide that plays an important role in antioxidant, anti-aging, and antiglycation activities, along with strong buffering properties; it is found in the muscles of most vertebrates and acts as a biomarker in chicken muscle [9,11,12].

### 3.2. Multivariate Statistical Analysis for Distinct Metabolic Fingerprints Reflecting Different Genotyping Traits of Chicken Breeds

In the current study, PCA and OPLS-DA models were constructed to visualize intrinsic and extrinsic similarities and differences of all groups and to investigate the metabolic differences of all groups and different classifications of chicken breasts, respectively. First, pairwise PCA models of TNC (100% Thai native), TSC (50% Thai native), and TNC crossbred (25% Thai native) compared with commercial broiler chicken (0% Thai native) were constructed (Figure 1A–D). All pairwise PCA score plots demonstrated the unique metabolic fingerprints of all breeds compared with commercial broiler chicken, of which the first principal component (PC1) of all models determined a clear class separation. Likewise, pairwise OPLS-DA score plots demonstrated distinct class discrimination (Figure 2A–D). OPLS-DA loadings plots further demonstrated that TNC crossbred, TSC, and TNC contained higher levels of anserine and anserine/carnosine compared with commercial broiler chicken (Figure 3A–D). Moreover, it is noteworthy that TNC (CH breed) and TSC showed higher levels of lactate than their commercial broiler counterpart did. These differences in metabolic traits could possibly result from the genetic improvement of each breed, with different purposes and selection indices reflecting the metabolic alteration; for example, commercial broiler chickens have been developed for faster growing, mainly selected on breast yield [31]. Other breeds have been developed for growth and egg production using conventional and molecular breeding tools, but they are still slower growing compared with commercial broilers because this genetic trait is limited and different criteria were used for genetic selection [32,33,34,35].

The comparison of metabolic differences within TNC (PD and CH), TSC, and TNC crossbred could not be distinguished among groups in either the PCA or OPLS-DA model (*p* > 0.05). The data are shown in Figure 4 for the pairwise comparison of CH versus KKU-ONE. The pairwise comparison model for PD versus KM showed similar results to the aforementioned model. TNC as CH versus TSC and TNC crossbred could not be distinguished among groups, possibly because TSC and TNC crossbred were developed by CH utilization to achieve a sustainable to developing four Thai synthetic chicken lines known as Khai Mook E-san, Kaen Thong, Soi Nin, and Soi Pet, whereas terminal cross known as KKU-ONE. Therefore, some genes may be close to one another [32,33,34,35]. This result showed that both TNCs and their crossbred forms have close potential.

### 3.3. Relative Concentration of Anserine and Anserine/Carnosine Content

Relative concentrations of anserine and anserine/carnosine content were obtained from the area under the peak of the ^1^H NMR spectra and obtained from chicken breast extracts of TNC crossbred, TSC, and TNC (PD and CH). All breeds demonstrated significantly higher contents of anserine and anserine/carnosine (*p* < 0.05) than commercial broiler chicken, as shown in Figure 5 and Table 2.

In the current study, TNC, TSC, and TNC crossbred extracts contained higher anserine and anserine/carnosine content than commercial broiler chicken did, and this result is in agreement with the previous findings of Liu et al., Kojima et al., Jayasena et al., Ali et al. [9,10,36,37] that native chicken meat contains higher anserine compared with commercial broiler chicken. The differences in anserine and carnosine among the chicken breeds may be attributed to the different muscle fiber types [9,19,38]. Previous reports [16,23,39] have suggested that native and crossbred chickens contained higher amounts of muscle fiber type IIB muscle than commercial broilers do. According to Jung et al. and Verdiglione et al. [19,40], chicken breast meat contains type IIB muscle fibers of the fast-twitch glycolytic type, which play an important role in anaerobic metabolism to adenosine triphosphatase (ATP). Normally, the accumulation of lactic acid is higher in the breast than it is in the thigh muscle in chickens [41]. Hence, chicken breast muscle needs endogenous compounds with high buffering properties, such as anserine and carnosine [19,42].

In addition, anserine content in chicken meat based on feed intake and protein level in the feed, refs. [43,44] reported that raised on different level of soybean meal has effect on histidine-containing chicken meat. In this study, all the birds were raised with a broiler diet with the starter feed containing 21% CP and growing feed consisting of 19% CP. In terms of the amount and kind of amino acid, we have no information because the company provided only feedstuff information. Nevertheless, we recorded the feed intake and found that total feed intake up to the time of slaughter of the TNC PD, TNC CH, TSC, TNC crossbred, and commercial broiler chickens were 4539.8, 3874.5, 3591.9, 3265.0, and 3439.1 g, respectively; there were no significant differences in the feed intake among breeds (*p* > 0.05). These results indicate that the differences in anserine and anserine/carnosine content may be genetic.

Among the chicken breeds, TNC—especially PD—and such Thai native crossbred chickens as KKU-ONE (Thai native genetic fraction to 25%) are popular in Thailand because of their greater growth performance and their texture. Many enterprises are interested in trying to take them up for modern trade, which requires component-quality nutrients/bioactive compounds that are beneficial to human health and wellbeing. From this perspective, the results of this study have the advantage of opening opportunities to enhance the competitiveness of the healthy food market and export abroad. Moreover, it can sustainably increase community income by developing and promoting functional food and natural products as a natural extract from native chicken meat and chicken broth for an aging society. Reference [45] suggested that anserine (beta-alanyl-3-methyl-L-histidine) supplementation improves memory functions in the context of Alzheimer’s disease in a mouse model, with a protective effect on the neurovascular units. In addition, ref. [13] reported that the lactate concentration in blood decreased and endurance performance improved after consuming chicken breasts, which is beneficial for people who like to exercise. Therefore, TNC and their crossbred forms have the potential to be a source of functional meat.

### 3.4. Antioxidant Activity of Dipeptide-Rich Chicken Breast Extracts

The DPPH radical scavenging assay is an accepted mechanism for screening radical scavenging activity [46]. In this study, the antioxidant activity of the extracts of breast meat was determined (Figure 6 and Table 3). The results showed that the antioxidant activity of TNC meat extract was higher than TNC crossbred and commercial broiler extracts were (*p* < 0.01). Comparing the CH breed (502.76^A^) with the other breeds, CH had the highest antioxidant activity when compared with PD (406.57^B^), KM (414.08^B^), KKU-ONE (421.71^B^), and broiler (294.86^C^) meat. References [9,47] have shown that native chicken meat extracts are rich in anserine and carnosine, with high antioxidant activity. Therefore, TNC, TSC, and TNC crossbred could potentially serve as a functional chicken meat source.

## 4. Conclusions

This study was conducted to identify the presence of dipeptide anserine and its antioxidant properties in chicken breast extract of Thai native and crossbred chickens compared with commercial broiler chickens. We found that the metabolites of Thai native and crossbred chickens could be distinguished from commercial broiler chicken in terms of their rich anserine and anserine/carnosine contents and higher antioxidant properties compared with commercial broiler chicken. Therefore, Thai native and crossbred chickens may have the potential to serve as functional meat sources.

## Figures and Tables

**Figure 1 animals-11-00902-f001:**
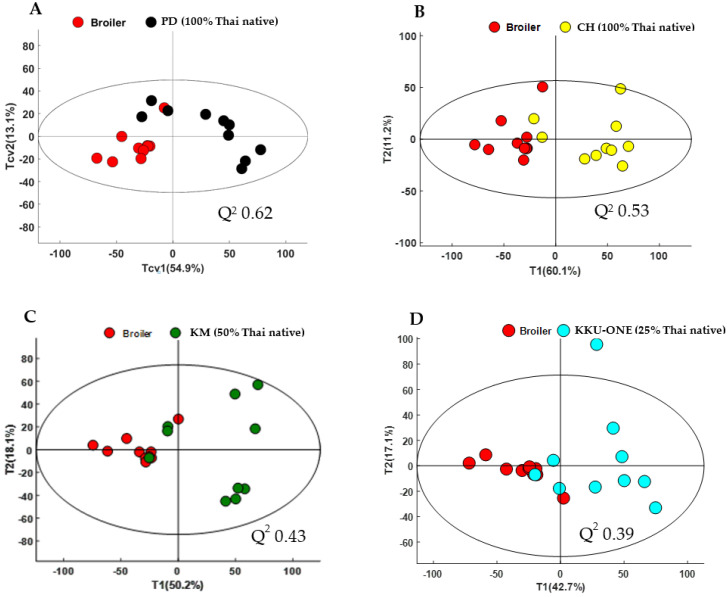
Principal component analysis (PCA) scores discriminating metabolites of chicken breast extracts of (**A**) commercial broilers versus PD = Pradu Hang Dam Mor Kor 55 (100% Thai native); Q^2^ = 0.62, (**B**) commercial broilers versus CH = Chee KKU 12 (100% Thai native); Q^2^ = 0.53, (**C**) commercial broilers versus KM = Khai Mook E-san (50% Thai native;); Q^2^ = 0.43, and (**D**) commercial broilers versus KKU-ONE (25% Thai native); Q^2^ = 0.39.

**Figure 2 animals-11-00902-f002:**
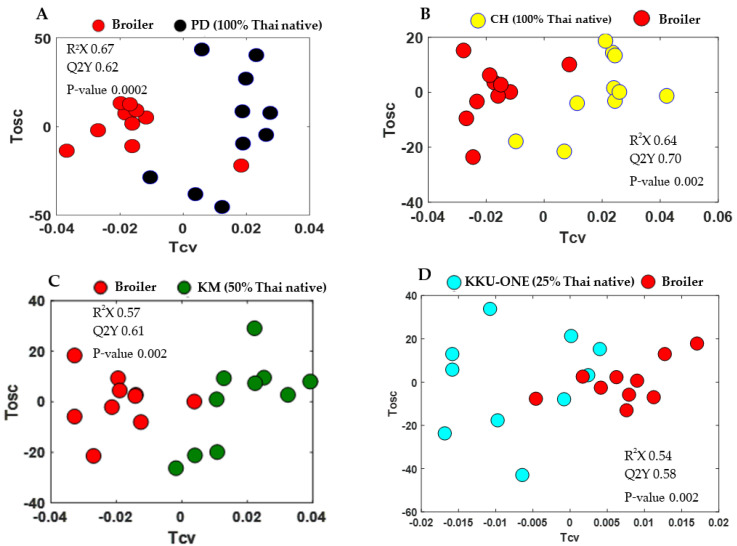
Orthogonal signal correction–Projection to Latent Structures–Discriminant Analysis (OPLS-DA) scores discriminating metabolites of chicken breast extracts of (**A**) commercial broilers versus PD = Pradu Hang Dam Mor Kor 55 (100% Thai native); R^2^X = 0.67, Q^2^Y = 0.62 and *p*-value = 0.0002, (**B**) commercial broilers versus CH = Chee KKU 12 (100% Thai native); R^2^X = 0.64, Q^2^Y = 0.70 and *p*-value 0.002, (**C**) commercial broilers versus KM = Khai Mook E-san (50% Thai native); R^2^X = 0.57, Q^2^Y = 0.61 and *p*-value = 0.002, and (**D**) commercial broilers versus KKU-ONE (25% Thai native); R^2^X = 0.54, Q^2^Y = 0.58 and *p*-value = 0.002.

**Figure 3 animals-11-00902-f003:**
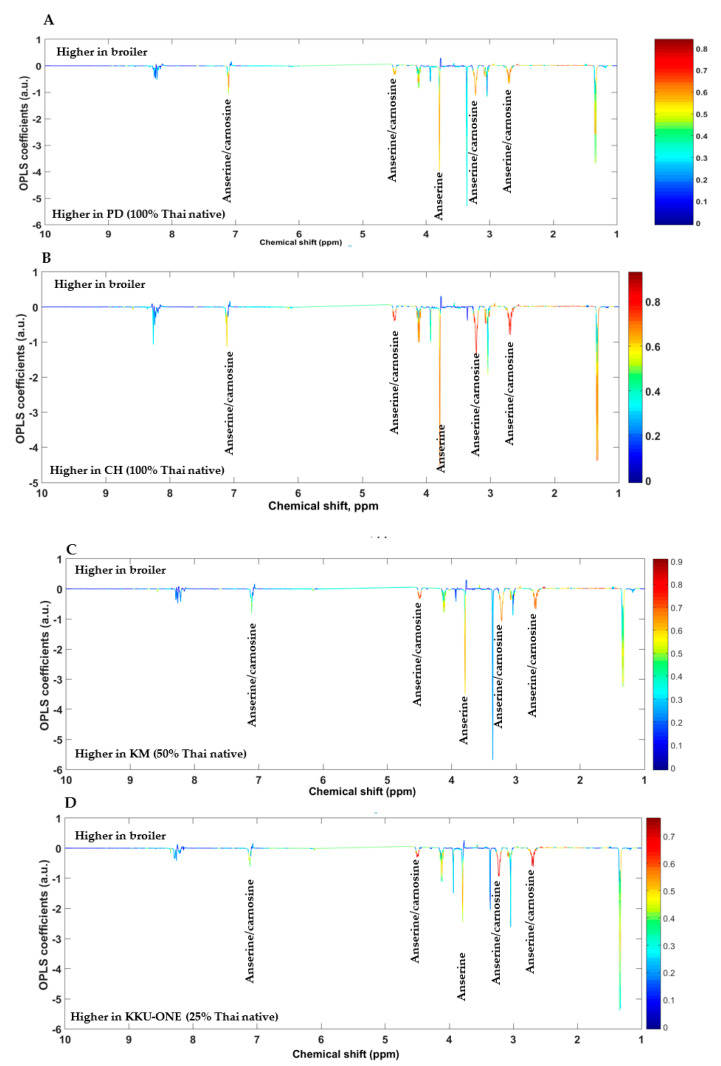
OPLS-DA loading plots discriminating metabolites of chicken breast extracts of (**A**) commercial broilers versus PD = Pradu Hang Dam Mor Kor 55 (100% Thai native), (**B**) commercial broilers versus CH = Chee KKU 12 (100% Thai native), (**C**) commercial broilers versus KM = Khai Mook E-san (50% Thai native), and (**D**) commercial broilers versus KKU-ONE (25% Thai native).

**Figure 4 animals-11-00902-f004:**
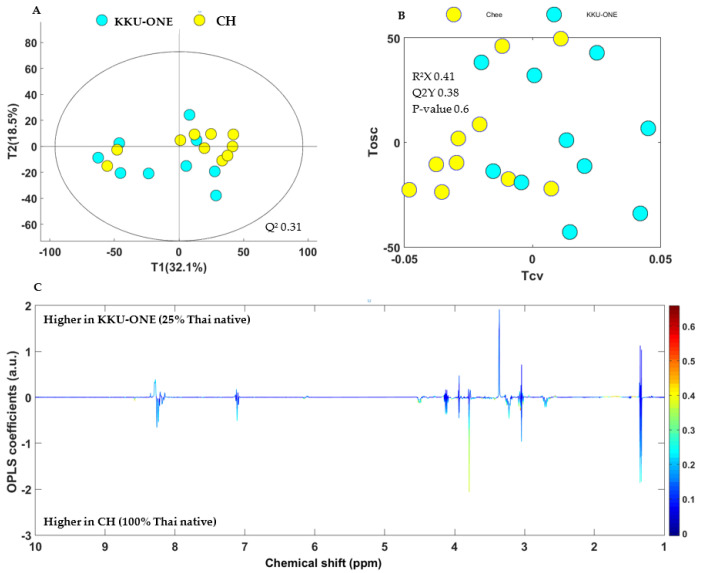
(**A**) PCA scores, (**B**) OPLS-DA scores, and (**C**) OPLS-DA loading plots of chicken breast extracts discriminating metabolites of KKU-ONE (25% Thai native) and CH = Chee KKU 12 (100% Thai native). R^2^X = 0.41, Q^2^Y = 0.38, *p*-value = 0.6.

**Figure 5 animals-11-00902-f005:**
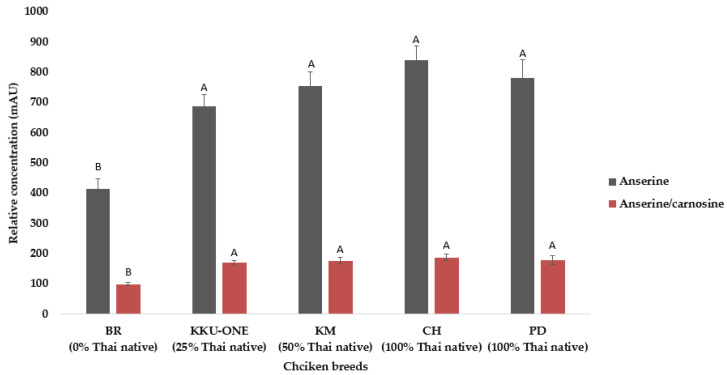
Relative concentrations of anserine (3.791 ppm) and anserine/carnosine (3.22 ppm) in chicken breast extracts by 400-MHz ^1^H-NMR spectra (0–10 ppm) from PD = Pradu Hang Dam Mor Kor 55 (100% Thai native), CH = Chee KKU 12 (100% Thai native), KM = Khai Mook E-san (50% Thai native), and KKU-ONE (25% Thai native). ^A,B^ represent the level of significance of the differences among the genotypes (*p* < 0.01).

**Figure 6 animals-11-00902-f006:**
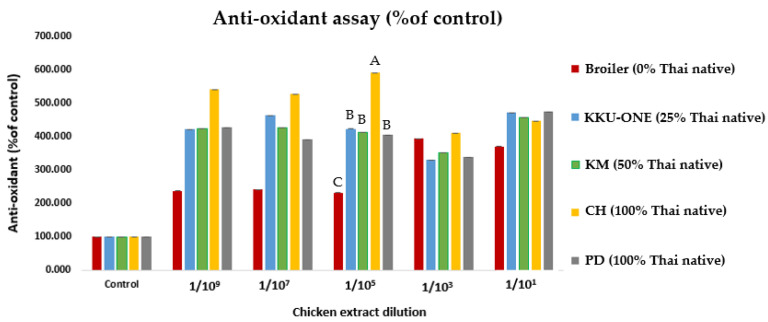
Effect of dipeptide-rich chicken extracts on antioxidant activity of PD = Pradu Hang Dam Mor Kor 55 (100% Thai native), CH = Chee KKU 12 (100% Thai native), KM = Khai Mook E-san (50% Thai native), and KKU-ONE (25% Thai native). ^A,B,C^ represent the level of significance of the differences among the genotypes (*p* < 0.01).

**Table 1 animals-11-00902-t001:** Metabolites and their values based on ^1^H NMR spectra.

Metabolite	ppm
Lactate	1.344 (d); 4.11 (q)
Alanine	1.48 (d); 3.79 (q)
Anserine/carnosine	2.69 (m); 3.22 (m); 4.51 (m); 7.11 (s)
Anserine	3.791 (s)
Creatine	3.0 (s); 3.93 (s)
Inositol monophosphate (IMP)	4.043 (s); 8.577 (s)
Inosine	3.042 (s):8.353 (s)

Keys: (s:) singlet; (d:); doublet; (t:) triplet; (q:) quartet; (m:) multiplet.

**Table 2 animals-11-00902-t002:** Relative concentrations of anserine and anserine/carnosine in chicken breast extracts.

Effects	BR	KKU-ONE	KM	CH	PD	SEM	*p*-Value
Breeds							
Anserine	404.37 ± 51.64 ^B^	691.71 ± 49.85 ^A^	757.58 ± 49.85 ^A^	837.50 ± 49.85 ^A^	780.64 ± 50.53 ^A^	50.34	<0.0001
Anserine/carnosine	100.21 ± 11.81 ^B^	169.93 ± 11.40 ^A^	177.01 ± 11.40 ^A^	187.85 ± 11.40 ^A^	179.20 ± 11.55 ^A^	11.51	<0.0001
Breed × sex							
Anserine							0.2394
Female	453.27 ± 54.41	777.02 ± 76.95	759.83 ± 76.95	811.37 ± 62.83	716.83 ± 88.85	72.00	
Male	253.76 ± 108.82	629.83 ± 62.83	751.06 ± 62.83	884.19 ± 76.95	807.97 ± 58.17	73.92	
Anserine/carnosine							0.1670
Female	105.65 ± 12.29	187.67 ± 17.39	177.06 ± 17.39	172.29 ± 14.19	162.52 ± 20.08	16.27	
Male	75.03 ± 24.59	158.48 ± 14.19	177.35 ± 14.19	210.61 ± 17.39	186.99 ± 13.14	16.70	

^A,B^ represent the level of significance of the differences among the genotypes (*p* < 0.01), PD = Pradu Hang Dam Mor Kor 55 (100% Thai native), CH = Chee KKU 12 (100% Thai native), KM = Khai Mook E-san (50% Thai native), and KKU-ONE (25% Thai native).

**Table 3 animals-11-00902-t003:** Antioxidant effect of dipeptide-rich chicken breast extract of TNC, TSC, and TNC crossbred animals.

Breeds	BR	KKU-ONE	KM	CH	PD	SEM
Antioxidant activity (%)	294.86 ^C^	421.71 ^B^	414.08 ^B^	502.76 ^A^	406.57 ^B^	84.83

^A,B,C^ represent the level of significance of the differences among the genotypes (*p* < 0.01). PD = Pradu Hang Dam Mor Kor 55 (100% Thai native), CH = Chee KKU 12 (100% Thai native), KM = Khai Mook E-san (50% Thai native), and KKU-ONE (25% Thai native) compared to BR = commercial broiler chicken.

## Data Availability

Additional data are available on request from the corresponding authors.

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
