# Peer review of "Thai Native Chicken as a Potential Functional Meat Source Rich in Anserine, Anserine/Carnosine, and Antioxidant Substances"

_animals, 2021, doi:10.3390/ani11030902_

Round 1
Reviewer 1 Report
The present manuscript entitled “Thai Native Chicken Rich in Anserine, Anserine/Carnosine, and antioxidant substances as Potential Functional Meat Source” by Charoensin et al. regards the evaluation of certain muscle metabolites in different chicken breeds/lines by means of NMR spectroscopy. Overall, the topic is of interest, although the international relevance is somehow limited. In general, linguistic aspects could be further improved. As regard the research, some concerns arose during the revision process. Unfortunately, various inaccuracies and missing information make this paper difficult to evaluate. Taken together, the overall quality of the current version is rather low. First, the authors should better contextualize the rationale behind this research and the importance that these results can have in terms of market opportunities for these breeds and/or potential consequences on human health (e.g. why the Authors performed the analysis on Thai Native Chickens? Do Thai Native Chickens have any higher capacity of synthesizing antioxidant compounds? Which is the actual diffusion and consumption of the meat of these breeds? What kind of compounds they are looking for?). In the introduction, there is very limited connection among the paragraphs and it should be revised and improved according to the information available in literature. Furthermore, the introduction is not very well focused and thus research hypothesis is not completely justified. Moreover, important aspects are missing in the M&Ms section (or at least they should be better explained) including the kind of NMR analysis (targeted or untargeted?), the criteria for bird processing (commercial broilers weight much more than 1.2-1.6 kg at 10 or 12 wks of age) and diet composition. Results presentation do not follow the statistical approach described in M&Ms and there is no indication regarding the statistical relevance of the main effects and their interaction. The relative concentration of each molecule in the meat samples should be reported accordingly. The results are only slightly and superficially discussed, resulting in a limited exploitation of such results. Unfortunately, the conclusion mirrors the discussion and thus it is not very accurate. In addition, Authors should provide much more evidence before declaring that Thai chicken crossbreeds could be marketed as functional chicken meat. For all these reasons, I recommend to reject the present manuscript, at least in its current form. However, as the topic is of interest, I’d suggest to the Authors to improve the overall quality of the manuscript and reconsider it as new submission.
Author Response
Firstly, I want to tell that I can not add line number in the revised manuscript so would highlight with yellow color into the sentence that I added and edit.
1.1 First, the authors should better contextualize the rationale behind this research
Answer: I have already edit the first paragraph of the introduction.
1.2 and the importance that these results can have in terms of market opportunities for these breeds and/or potential consequences on human health (e.g. why the Authors performed the analysis on Thai Native Chickens? Do Thai Native Chickens have any higher capacity of synthesizing antioxidant compounds? Which is the actual diffusion and consumption of the meat of these breeds? What kind of compounds they are looking for?).
Answer: I have already edited and added information in first and second paragraph of introduction. And I have already added information in the topic 3.5 Market opportunities for these breeds and/or potential consequences on human health
2. In the introduction, there is very limited connection among the paragraphs and it should be revised and improved according to the information available in literature. Furthermore, the introduction is not very well focused and thus research hypothesis is not completely justified.
2.1 Introduction, there is very limited connection among the paragraphs
Answer: I have already edited the introduction
2.2 Introduction is not very well focused and thus research hypothesis is not completely justified
Answer: I have already edit the research hypothesis in the introduction
3. Moreover, important aspects are missing in the M&Ms section (or at least they should be better explained) including the kind of NMR analysis (targeted or untargeted?), the criteria for bird processing (commercial broilers weight much more than 1.2-1.6 kg at 10 or 12 wks of age) and diet composition.
3.1 Important aspects are missing in the M&Ms section (or at least they should be better explained)
Answer: I I have already edited and put details in materials and methods
3.2 The criteria for bird processing (commercial broilers weight much more than 1.2-1.6 kg at 10 or 12 wks of age)
Answer: this bodyweight is the marketing weight of Thailand for Thai native and Thai native crossbred, Thai consumers consume about 1.2-1.6 kg (live weight) in terms of commercial broiler almost we consume retail cut in the supermarket they collected about 2.0-2.2 kg. I have already added the information in the M&Ms section, second paragraph of topic 2.1. with blue highlight.
3.3 Diet composition
Answer: I have already added the information in the Materials and Methods, the first paragraph of topic 2.1 with a green highlight.
4.1 Results presentation do not follow the statistical approach described in M&Ms and there is no indication regarding the statistical relevance of the main effects and their interaction
Answer: I have already edited result presentation in the topic 3.1, 3.2, 3.3 and 3.4 and represented follow the statistical approach described in M&Ms and added the statistical relevance
4.2 The relative concentration of each molecule in the meat samples should be reported accordingly
Answer: I focused on anserine, I have already added relative concentration of anserine and anserine/carnosine in result the topic 3.3
5. The results are only slightly and superficially discussed, resulting in a limited exploitation of such results. Unfortunately, the conclusion mirrors the discussion and thus it is not very accurate.
Answer: I have already edited the result in the topic 3.1, 3.2, 3.3, 3.4 and 3.5 included discussion and conclusion
6. In addition, Authors should provide much more evidence before declaring that Thai chicken crossbreeds could be marketed as functional chicken meat. For all these reasons,
Answer: I have already edited and added information in the result topic 3.1, 3.2, 3.3, 3.4 and 3.5
7. I recommend to reject the present manuscript, at least in its current form. However, as the topic is of interest, I’d suggest to the Authors to improve the overall quality of the manuscript and reconsider it as new submission.
Answer: I have already edit overall manuscript follow the comment of reviewer.

Reviewer 2 Report
The results and data in this paper represent a significant contribution to knowledge in potential functional meat source for chicken and are to merit publication. Hence the paper is accepted.
Author Response
Thank you very much
May I sent a revised manuscript followed by another reviewer
Reviewer 3 Report
The manuscript is interesting. The analytical tests on the samples are conscientious and informative.
The following questions arise from the material and method:
- The potential bioactive compounds are formed in the body from precursors for example essential amino acids. Since these amino acids come from the poultry feed, it is necessary to specify the feed composition. Furthermore, information about the amino acids that are considered to be precursors must be given.
- The feed intake of the chickens of the different breeds will be different. Therefore at least the total feed intake up to the time of slaughter of the chickens must be stated.
- The feed intake should be used to calculate how high the intake from the preliminary stages (amino acids) was.
- These results are to be discussed and compared with literature.
- From this, conclusions can be drawn for the supply of the chickens with the preliminary stages (amino acids).
Author Response
- The potential bioactive compounds are formed in the body from precursors for example essential amino acids. Since these amino acids come from the poultry feed, it is necessary to specify the feed composition. Furthermore, information about the amino acids that are considered to be precursors must be given.
Answer: we have already added feed composition in topic 2.1 with blue highlight
- The feed intake of the chickens of the different breeds will be different. Therefore at least the total feed intake up to the time of slaughter of the chickens must be stated.
Answer: we have already added feed intake up to the time of slaughter in the result topic 3.3 with a yellow highlight.
- The feed intake should be used to calculate how high the intake from the preliminary stages (amino acids) was.
Answer: We propose which breed has the potential to be functional meat under controlling the feed is same was raised in all of the breed to know their real genetic along with synthesis of bioactive compounds as anserine and so on.
- These results are to be discussed and compared with the literature.
Answer: For the result, I have already added information for discussed and compared with the literature in the result the third paragraph of topic 3.3 with a yellow highlight.
- From this, conclusions can be drawn for the supply of the chickens with the preliminary stages (amino acids).
Answer: We propose on their genetic under same diet composition, however, we have already added information feed intake in the result the third paragraph of topic 3.3 with a yellow highlight.

Round 2
Reviewer 1 Report
I sincerely appreciate the efforts put by the Authors to amend the manuscript. Unfortunately, I think that the manuscript is still not acceptable for publication on Animals. First, linguistic aspects need to be deeply and thoroughly reviewed. The introduction is still not focused and somehow confusing. The rationale and the justification for the study need to be better presented. Results presentation do not follow the statistical approach described in M&Ms (e.g. the gender effect should be reported for all the parameters, the use of acronym is confusing) and there is no indication regarding the statistical relevance of the main effects and their interactions. The new section (“3.5 Market opportunities for these breeds and/or potential consequences on human health”) should be integrated either in the introduction or in the discussion/data presentation. The conclusions are still not accurate and somehow speculative. Overall, I’d suggest to the Authors to reconsider the manuscript for a new submission after a rigorous revision process which includes both linguistic and scientific aspects.
Author Response
1. First, linguistic aspects need to be deeply and thoroughly reviewed.
Answer: We have already checked and edited for language and sent to the company proof by editors on the academic service are capable.
2. The introduction is still not focused and somehow confusing.
Answer: We have already added information and edited.
3. The rationale and the justification for the study need to be better presented.
Answer: We have already edited and added information.
4. Results presentation do not follow the statistical approach described in M&Ms (e.g. the gender effect should be reported for all the parameters, the use of acronym is confusing)
Answer: We have already edited the presentation of the results followed the statistical approach described and some parameter we were mixed sexes because of previous results showing that interactions between breed and sex have no significance and we have already added information in topic 2.3.1 and mark yellow highlight. In terms of acronym, I have already edited.
5. and there is no indication regarding the statistical relevance of the main effects and their interactions.
Answer: We have already edited in table 2.
6. The new section (“3.5 Market opportunities for these breeds and/or potential consequences on human health”) should be integrated either in the introduction or in the discussion/data presentation.
Answer: We have already edited and integrated into the introduction and discussion/data presentation.
7. The conclusions are still not accurate and somehow speculative.
Answer: We have already edited.
8. Overall, I’d suggest to the Authors to reconsider the manuscript for a new submission after a rigorous revision process which includes both linguistic and scientific aspects.
Answer: We have already edited all followed reviewer’s comments.
